# Control of Replication Stress Response by Cytosolic Fe-S Cluster Assembly (CIA) Machinery

**DOI:** 10.3390/cells14060442

**Published:** 2025-03-16

**Authors:** Chiara Frigerio, Michela Galli, Sara Castelli, Aurora Da Prada, Michela Clerici

**Affiliations:** Dipartimento di Biotecnologie e Bioscienze, Università degli Studi di Milano-Bicocca, 20126 Milano, Italy; c.frigerio32@campus.unimib.it (C.F.); m.galli50@campus.unimib.it (M.G.); s.castelli30@campus.unimib.it (S.C.); a.daprada@campus.unimib.it (A.D.P.)

**Keywords:** DNA replication stress, iron–sulfur cluster, cytosolic Fe-S cluster assembly (CIA), Mec1/ATR, DNA polymerases, dNTPs, DNA nucleases

## Abstract

Accurate DNA replication is essential for the maintenance of genome stability and the generation of healthy offspring. When DNA replication is challenged, signals accumulate at blocked replication forks that elicit a multifaceted cellular response, orchestrating DNA replication, DNA repair and cell cycle progression. This replication stress response promotes the recovery of DNA replication, maintaining chromosome integrity and preventing mutations. Defects in this response are linked to heightened genetic instability, which contributes to tumorigenesis and genetic disorders. Iron–sulfur (Fe-S) clusters are emerging as important cofactors in supporting the response to replication stress. These clusters are assembled and delivered to target proteins that function in the cytosol and nucleus via the conserved cytosolic Fe-S cluster assembly (CIA) machinery and the CIA targeting complex. This review summarizes recent advances in understanding the structure and function of the CIA machinery in yeast and mammals, emphasizing the critical role of Fe-S clusters in the replication stress response.

## 1. Introduction

DNA replication is crucial for all living organisms to transmit genetic information to their progeny. During DNA replication, genome integrity is challenged by both endogenous and exogenous factors, including polymerase-blocking DNA lesions, DNA secondary structures, DNA-binding proteins and DNA-RNA hybrids, all of which can compromise the faithful transmission of genetic material to offspring [1,2]. In addition, alterations in origin firing activation, the depletion of replication factors or low deoxyribonucleotide triphosphate (dNTP) levels can induce replication stress, which hinders replication fork progression [3,4]. Both DNA damage and replication stress can lead to genome instability, which is a hallmark of neurodegenerative diseases and pre-neoplastic lesions, as well as a driver of carcinogenesis [5,6].

Iron–sulfur (Fe–S) clusters are increasingly recognized as essential cofactors in the cellular response to replication stress. They have been identified in various enzymes crucial for DNA metabolism in both yeast and humans, including DNA polymerases, DNA helicases and DNA repair enzymes [7]. While the precise functions of these clusters are not yet fully understood for all Fe–S-containing proteins, they are known to frequently act as enzymatic cofactors and play a role in sensing and responding to the redox conditions within the cell. In some cases, they can also contribute to the structural stability of protein complexes [7,8].

Baker’s yeast has been widely used as a powerful model system to dissect the molecular mechanisms of the replication stress response and to study Fe–S cluster biogenesis in living cells, including Fe-S cluster transfer to target proteins and their impact on protein activity. Here, we review recent advancements in understanding the role of Fe–S clusters in modulating the cellular response to replication stress. Our focus is primarily on findings obtained in *Saccharomyces cerevisiae*, but we also highlight both similarities and differences between yeast and mammals. Mutations in genes encoding proteins involved in the replication stress response, as well as in factors required for Fe–S cluster biogenesis, have been identified in various cancers and genetic disorders. Therefore, investigating these pathways and their interplays could provide valuable insights into disease pathogenesis and cancer development, potentially guiding new therapeutic strategies.

## 2. The Cellular Response to Replication Stress

### 2.1. DNA Replication and DNA Polymerases

DNA replication requires several proteins, including DNA helicases and DNA polymerases, which are assembled into a structure called the replisome, containing all the activities necessary to copy DNA strands with high fidelity [9]. The replisome contains the Cdc45/Mcm2-7-GINS (CMG) replicative helicase that unwinds the parental chromosomes, generating DNA replication forks. At the replication forks, the DNA polymerases ε (Pol ε) and δ (Pol δ) catalyze DNA synthesis on the leading and lagging strands, respectively. Nevertheless, these DNA polymerases are not able to start de novo synthesis, for which they need the action of the Pol α-DNA primase complex that synthesizes RNA-DNA primers with a free 3’ hydroxyl group. The Pol α-primase complex is connected to the replisome, and specifically to the GINS complex, via Ctf4 in both yeast and humans (Figure 1) [10,11,12,13].

Pol δ is composed of the catalytic subunit Pol3/p125 and the accessory subunits Pol31 and Pol32 in yeast, and p50, p66 and p12 in humans [8]. Yeast Pol3 has a proofreading exonuclease activity, which belongs also to mammalian p125, and mutations in *POL3* increase the mutation rate [14]. In addition, Pol3 binds a Fe-S cluster, which promotes its association with Pol31 and Pol32 and the formation of the active complex [15]. Pol ε is composed in yeast of the catalytic subunit Pol2 and the accessory subunits Dpb2, Dpb3 and Dpb4 [16,17]. The C-terminal domain (CTD) of Pol2 binds Dpb2, which targets the polymerase toward the replisome [18], whereas Dpb3 and Dpb4 interact with the central portion of Pol2 and participate in DNA binding and chromatin remodeling [19]. In human cells, Pol ε is composed of the catalytic subunit p261 and the accessory subunits p59, p17 and p12 [8]. Pol α is present in eukaryotic cells in tight association with the DNA primase complex. Pol α is composed of the catalytic subunit Pol1/p180 and the accessory subunit Pol12/p70, while the primase complex is composed of Pri1/p49, with catalytic activity, and the accessory subunit Pri2/p58, which contains a Fe-S cluster in the CTD [20,21]. The primase complex synthesizes the RNA primer, whereas Pol1/p180 elongates the RNA primer by adding dNTPs, thus creating a suitable substrate for the action of Pol ε and Pol δ.

### 2.2. Replication Stress and the Replication Checkpoint

Conditions known as replication stress, which include the presence of DNA-binding proteins, depletion of replication factors or low dNTP levels, hamper the progression of DNA replication (Figure 1) [3,4]. Dysfunctional replication forks are unable to complete DNA synthesis, and cells that are unable to complete DNA replication can accumulate DNA lesions and/or chromosome rearrangements [22]. Cells cope with replication stress by activating the S phase checkpoint, a complex cellular response that couples a delay in S phase progression with DNA repair and the recovery of DNA replication [23]. This response includes the regulation of replication origin firing and of fork architecture, as well as a reorganization of the replication machinery that depends on the source of stress [4,24]. On one hand, small DNA lesions that pass through the replisome and block DNA replication could result in a physical uncoupling between replicative helicases and DNA polymerases, generating a single-stranded DNA (ssDNA) region that is coated by Replication Protein A (RPA) [25]. On the other hand, bulky impediments, such as protein–DNA complexes, interstrand cross-links or DNA adducts, hinder the progression of replication forks by blocking the unwinding of parental DNA strands, generating a stalled replication fork [1,2]. In addition, most cytotoxic lesions such as DNA double-strand breaks (DSBs) can occur when replication forks encounter DNA lesions or ssDNA gaps, generating one-ended DSBs [26].

The presence of either long ssDNA stretches or DSBs activates the checkpoint. The checkpoint is initiated by the protein kinases Mec1/ATR and Tel1/ATM, which recognize long ssDNA stretches and DSBs, respectively [27]. In turn, Mec1/ATR and Tel1/ATM block cell cycle progression through the phosphorylation of the downstream checkpoint kinases Rad53/CHK2 and Chk1/CHK1, whose activation requires the action of the mediators Mrc1/CLASPIN or Rad9/53BP1 [27]. During S phase, Mec1/ATR recognizes ssDNA covered by RPA, which is a common intermediate in many physiological pathways such as DNA replication and transcription. Obstacles to replication fork progression cause fork stalling and the generation of long ssDNA stretches due to nucleolytic processing and the uncoupling between the replicative helicase and polymerase (Figure 1) [2]. Mec1/ATR activation during S phase requires higher amounts of RPA-coated ssDNA compared to what is needed to trigger checkpoint activation in different cell cycle phases, such as G1 or G2. This suggests the presence of a threshold for checkpoint activation during S phase, ensuring that the ssDNA typically produced at functional replication forks is not sufficient to induce a DNA damage response [28]. The recognition of RPA-coated ssDNA by Mec1/ATR depends on the Mec1/ATR-interacting protein, called Ddc2/ATRIP, but the full activation of the kinase also requires the 9-1-1 complex (known also as PCNA-like), which is formed by Ddc1, Rad17 and Mec3 in yeast and RAD9, RAD1 and HUS1 in humans. This complex is loaded on the DNA by Rad24-Rfc2-5 and activates Mec1/ATR, recruiting Dpb11/TopBP1 [29]. The activation of Rad53/CHK2 by Mec1/ATR in response to replication stress is not mediated by the DNA damage checkpoint protein Rad9/53BP1 but requires the mediator Mrc1/CLASPIN. Mrc1/CLASPIN is a component of the replisome; it interacts with both DNA Pol ε and Mcm6 and forms a complex with Tof1/TIMELESS and Csm3/TIPIN, which contributes to the stabilization of the replisome on DNA [30]. Mec1/ATR phosphorylates Mrc1/CLASPIN, which in turn activates Rad53/CHK2. Chk1 in yeast has a relatively minor role in the S phase checkpoint and differs from mammalian CHK1, which is known as the functional counterpart of yeast Rad53 [29]. Upon activation, the effector kinases Rad53 and CHK1 regulate several processes in order to fix DNA damage. These processes include the inhibition of late origin firing, the arrest of the cell cycle, the regulation of dNTP production, the promotion of DNA damage response gene transcription and the remodeling of the replication forks (Figure 1) [27,29]. The suppression of late origin firing and the inhibition of cell cycle progression supply extra time for the resolution of stress and allow the cell to complete DNA replication in the proximity of blocked replication forks. In humans, CHK1 negatively regulates CDK-dependent phosphorylation events at replication origins, blocking the loading of pre-initiation complex factors [31], while in yeast, Rad53 phosphorylates and inactivates the initiation factors Sld3 and Dbf4, thus blocking origin firing [32]. Checkpoint activation also prevents the entry of cells with under-replicated or damaged DNA into mitosis. In fact, CHK1 phosphorylates and inactivates CDC25, preventing G2-M transition. Metaphase–anaphase transition in yeast is instead inhibited by stabilizing the securin Pds1 [29].

### 2.3. The Control of dNTPs by the Replication Checkpoint

The homeostasis of the dNTP pool depends on the coordinated functions of three main pathways—de novo biosynthesis, nucleoside salvage and nucleotide catabolism—and is tightly regulated by complex feedback networks [33,34,35]. The enzyme ribonucleotide reductase (RNR) catalyzes the rate-limiting step in de novo biosynthesis pathways, which are the main pathways involved in dNTP production in *S. cerevisiae*, although a pyrimidine deoxyribonucleotide salvage pathway has also been described. This makes yeast an excellent model system for studying de novo dNTP synthesis as well as RNR function and regulation.

RNR catalyzes the reduction of ribonucleoside diphosphates (rNDPs) to deoxyribonucleoside diphosphates (dNDPs), the first step in dNTP biosynthesis. Its specific activity on different rNDPs is subject to sophisticated allosteric effector controls that are crucial to maintain a balanced dNTP pool. This control is ensured by two allosteric control sites located into the R1 subunit, which also contains the catalytic site [33,34]. In eukaryotes, RNR is a class I heterotetrameric enzyme, which comprises a large R1 subunit, composed of a homodimer of α peptides (α2), named Rnr1 in *S. cerevisiae* and RRM1 in mammals, and a small R2 subunit, which is composed of two identical β peptides (RRM2) in mammals (β2) and of two different proteins denoted Rnr2 and Rnr4 in yeast (β-β’) [33,34]. Yeast cells also express a second RNR large subunit, called Rnr3. Rnr3 is expressed at very low levels compared to the other three RNR subunits and is not essential for cell viability [36,37]. Despite Rnr3 exhibiting low endogenous activity alone, its activity increases in association with Rnr1, and *RNR3* overexpression rescues the lethality of *rnr1* null mutants [36,37]. The low levels of Rnr3 are due to the repression of *RNR3* transcription by the concerted action of the chromatin remodeler ISW2 and the Ssn6-Tup1 repression complex, which are recruited to *RNR2*, *RNR3* and *RNR4* promoters by the DNA-binding protein Crt1 [38]. An additional RNR repression mechanism is mediated by the protein Sml1, which binds the yeast R1 subunit and inhibits RNR activity when DNA synthesis is not required [39]. Moreover, an alternative R2 subunit (RRM2B or p53R2) is expressed under the control of p53 in humans and has important functions in regulating mitochondrial DNA replication [34]. *RRM2B/p53R2* transcription is also induced in response to DNA damage. RRM2B/p53R2 forms an active RNR holoenzyme in association with RRM1, which supports DNA repair [29]. Once DNA replication and/or repair is completed, RRM2/p53R2 undergoes degradation through ubiquitin-dependent mechanisms involving two E3 ubiquitin ligase complexes: the Skp1/Cullin/F-box (SCF) complex and the anaphase-promoting complex (APC) [40].

In response to DNA damage and replication stress, RNR expression and activity are induced to upregulate the dNTP pool. Yeast Mec1 activates Rad53, which in turn activates the Dun1 effector kinase by phosphorylation. Then, Dun1 enhances RNR function by phosphorylating and inhibiting Crt1 and Sml1 [38,41]. Mec1 and Rad53 regulate the dNTP pool even in the absence of DNA damage, and this function is essential for cell viability. In fact, cells lacking Mec1 or Rad53 are unviable, but this lethality can be rescued by increasing the dNTP pool through the deletion of *SML1* or the overexpression of *RNR* genes [39].

### 2.4. Fork Remodeling After Replication Stress

Eukaryotic cells have developed sophisticated mechanisms to rescue nonfunctional replication forks and promote fork recovery. These mechanisms depend on the type of obstacle and upon the organization of the arrested forks. Arrested forks can restart by merging forks emanating from dormant origins. This type of origin is abundant in the cell, and it is activated only in the presence of replication stress [42]. Replication forks that are unable to restart from dormant origins could collapse for the release of the replisome from DNA, thus generating structures recognized as DNA damage. Moreover, replication forks coming across DNA lesions can skip them using several strategies with minimal effect on DNA synthesis [43]. Replisomes could omit the lesion through repriming mechanisms that restart DNA synthesis downstream of the lesion [44]. Indeed, replication fork progression could be allowed by DNA polymerases, known as translesion DNA polymerases, which copy the genetic information through a lesioned DNA strand, with lower accuracy compared to the canonical DNA polymerases [45]. Alternatively, the rescue of the replication fork after repriming could involve recombination mechanisms, such as template switching, which requires the action of recombination factors to restore normal DNA replication [24]. An additional key mechanism for stabilizing and restarting replication forks includes the paring of newly synthesized DNA strands and the creation of a four-way structure like a Holliday junction (HJ) [46]. This process, known as fork reversal, was first recognized in yeast as a detrimental conversion that occurs when the Mec1/ATR pathway is abrogated [47]. However, reversed forks have been detected in different species in response to genotoxic agents and have been proposed to stabilize replication forks during the replication stress response. Fork reversal and the subsequent recovery of the reversed fork involve the action of several factors such as nucleases, helicases and recombination factors [24]. For example, the recombinase RAD51 facilitates fork reversal by promoting the coordinated paring of the two newly synthesized DNA strands [48], while the human RECQ1 helicase is essential for the subsequent fork restart [24]. Moreover, different DNA translocases, such as Rad5 in yeast, may also play a role in promoting fork reversal, although their precise function must be deeply investigated [49]. Moreover, RAD51 is involved in the restart and repair of blocked or broken replication forks through a process called HR-mediated fork restart. This mechanism is applied to forks that encounter replication-associated one-ended DSBs or uncoupled or reversed forks after the cleavage of one strand by structure-specific endonucleases such as Mus81 [50].

## 3. The Iron–Sulfur Machinery

### 3.1. Structure and Function of Fe-S Clusters

Fe-S clusters are present in proteins that regulate a wide array of cellular functions, including DNA metabolism, mitochondrial respiration, lipid metabolism and protein translation [51]. Fe-S clusters are ancient, small, inorganic protein cofactors composed of ferrous (Fe^2+^) or ferric (Fe^3+^) iron and sulfur inorganic ions (S^2-^). Some Fe-S proteins can also contain additional heavy metals such as molybdenum, vanadium or nickel [52]. Fe-S clusters can be found in three different forms: [2Fe-2S]^+^, [4Fe-4S]^2+^ and [3Fe-4S]^+^. In the simplest form [2Fe-2S]^+^, sulfur and iron ions coordinate four cysteine side chains, thus creating a rhombic structure. [4Fe-4S]^2+^ is considered a duplication of the [2Fe-2S]^+^ structure, where iron and sulfur alternately occupy the corners of the cube defined by ion disposition. [3Fe-4S]^+^ is the rarest form; it has a cuboidal structure in which one corner of the cube is unoccupied. The most common coordination partner for the iron ions is the sulfur atom of cysteine residues. However, histidine, and in rare cases arginine, serine, peptidyl-N and non-protein ligands, may also be used [53,54,55].

No single common consensus motif for binding a Fe-S cluster has been identified. However, some Fe-S proteins exhibit particular sequences, such as the CX_4_CX_2_CX ≈ 30C or the CX_2_CX_2_CX_20–40_ motifs. There is a slight preference for proline and tyrosine/phenylalanine residues in the first and second positions after the cysteine, respectively [56]. In addition, a Lys-Tyr-Arg (LYR)-like motif was identified in different Fe-S proteins and was found to mediate the interaction of the HSC20 chaperone with Fe-S client factors involved in the respiratory chain in human cells [57]. The LYR motif is also present in mitochondrial Fe-S proteins of some yeasts, leading to the suggestion that it could drive the insertion of Fe-S clusters in subclasses of Fe-S client proteins [57]. Despite the existence of these Fe-S consensus sequences, many Fe-S proteins do not follow these patterns. This inconsistency makes it challenging to predict and identify Fe-S proteins solely based on their primary sequence information. To address this challenge, several tools have been developed in order to identify novel possible Fe-S proteins [58,59,60].

Fe-S clusters serve various functions, although for some Fe-S proteins, the molecular role of the cofactors remains enigmatic [60]. In certain cases, these clusters can accept electrons from one partner and donate them to other molecules due to the ability of iron ions to switch between reduced and oxidized states. This electron transfer capability is exploited in enzymes involved in nitrogen fixation, photosynthesis and mitochondrial or bacterial respiration [53,61,62]. Fe-S proteins also play roles in non-redox processes: for example, they can function as a Lewis-acid catalyst, they are engaged in radical-mediated catalysis and they can serve as sulfur donors for the synthesis of cofactors [7,61]. In addition, the instability of Fe-S clusters in response to varying iron and oxygen levels makes them suitable for sensing ambient oxygen, gases and iron concentrations, hence regulating gene expression in response to environmental changes [63]. Fe-S clusters have been proposed to be important for the structure and activity of Fe-S proteins involved in DNA metabolism, as well as for their ability to bind DNA and detect DNA damage [41,64]. Finally, in some cases, Fe-S clusters may simply serve a structural role, stabilizing the rest of the polypeptide chain [53].

### 3.2. Biogenesis of Mitochondrial, Cytosolic and Nuclear Fe-S Cluster Proteins

Fe-S clusters can spontaneously assemble on purified proteins in vitro from ferrous iron and sulfide. However, in living cells, the biogenesis of Fe-S clusters requires complex biochemical assembly systems [7,52]. The biogenesis of Fe-S clusters requires cooperation between the mitochondrial Fe-S cluster (ISC) system and the cytosolic Fe-S protein assembly (CIA) system. The ISC system generates clusters for mitochondrial Fe-S proteins and a sulfur-containing compound that serves in the maturation of cytosolic and nuclear Fe-S proteins. This compound is delivered to the cytosol, where the CIA machinery assembles the Fe-S cluster and inserts it into the target proteins (Figure 2). ISC and CIA factors are conserved in eukaryotes, and most of them are essential for cell viability (Table 1) [7,61].

The ISC pathway starts with the delivery of iron and sulfur ions to the scaffold protein Isu/ISCU. Sulfur is released by the cysteine desulfurase Nfs1 from cysteine residues, while the required electrons are transferred from NADH through the electron transfer chain formed by the ferredoxin Yah1/FDX2 and the ferredoxin reductase Arh1/FDXR [52,53,65,66]. The source of iron is less clear; it is probably imported in the mitochondrion and then delivered to Isu/ISCU thanks to Yfh1/frataxin [53,67,68]. This peptide functions as an iron donor for Fe-S cluster synthesis in vitro, but concrete in vivo evidence of frataxin supplying iron ions is still lacking. Recent data suggest that it also functions as an allosteric regulator, thus increasing the cluster formation rate [54,69]. The ISC pathway also involves chaperones such as Ssq1/HSPA9 and Jac1/HSC20, which stabilize the scaffold protein and facilitate the release of the cluster [70]. The complex formed by ISCA1, ISCA2 and IBA57 assembles a [4Fe-4S]^2+^ cluster by the reductive coupling of two [2Fe-2S]^+^ clusters. This complex is also conserved in yeast, where ISCA1 and ISCA2 are called Isa1 and Isa2, respectively. How these three proteins physically and functionally interact with each other is still unclear [54,55,69]. Finally, NFU1/Nfu1 promotes the binding of the cluster to the target mitochondrial apoproteins [54,69,71].

Then, the ISC pathway exports a sulfur-containing compound of still-unknown identity from the mitochondrion to the cytosol for the biogenesis of cytosolic Fe-S clusters (Figure 2). The ABC transporter ABCB7 (Atm1 in yeast) is involved in this export [72]. However, in mammalian cells, the presence in the cytosol of small amounts of several ISC components suggests that maybe Fe-S clusters are not assembled exclusively in the mitochondrion. This aspect is still debated [69].

Cytosolic Fe-S cluster formation begins with the two P-loop NTPases Nbp35/NUBP1 and Cfd1/NUBP2, which form a heterotetrameric protein complex that coordinates different [4Fe-4S]^2+^ clusters [7,73]. Similarly to the role of ferredoxin in the mitochondrial process, the cytosolic pathway requires an electron transfer chain which involves the diflavin reductase protein Tah18/NDOR1 and the Fe-S protein Dre2/CIAPIN1 (Figure 2) [61,74]. Biogenesis further requires the cytosolic monothiol glutaredoxin Grx3-Grx4 (human PICOT), which promotes Fe-S cluster assembly on Dre2 [75,76]. The second major step of cytosolic Fe-S protein biogenesis involves the release of the newly assembled [4Fe-4S]^2+^ cluster from Cfd1-Nbp35 and its transfer to the late-acting CIA proteins. This reaction is mediated by Nar1/IOP1, which contains C- and N-terminal motifs with four cysteines each coordinating two [4Fe-4S]^2+^ clusters. Nar1/IOP1 functions as an adapter at the interface between early (Cfd1-Nbp35, Tah18-Dre2) and late CIA [7]. The late-acting CIA proteins form the CIA targeting complex (Cia1-Cia2-Met18 in *S. cerevisiae* and CIA1-CIA2B-MMS19 in mammals) and are responsible for the insertion of the cluster into the target proteins (Figure 2). We will now focus on the structure and function of the proteins that form the CIA targeting complex. For further details on the other steps of Fe-S cluster biogenesis, we refer the reader to excellent reviews [54,55,69,77,78].

### 3.3. The CIA Targeting Complex: Structure and Activity

The CIA targeting complex includes Cia1/CIA1, Cia2/CIA2B and Met18/MMS19 subunits [61]. These components physically interact with a large number of target proteins in the cytoplasm and nucleus. The CIA targeting complex is proposed to mediate apo-target recognition. However, the specific function(s) of the individual subunits within this complex has/have not been clearly defined [79].

#### 3.3.1. Cia1

Cia1 belongs to a large protein family characterized by WD40 repeat domains that form a β-propeller structure with seven pseudo-symmetrically orientated blades around a central axis. This geometry serves as a docking site for Fe-S proteins [80]. Cia1 binds Cia2, while Met18-Cia1 complexes have not been observed. Thus, Cia1’s interaction platform is probably used for target recognition or for the binding of Nar1 [81]. Cia1 arginine 127 in yeast is important for protein–protein interactions because its mutation in glutamic acid (*cia1-R127E*) affects cell growth and CIA machinery function [80]. The human homologous CIA1 was proposed to also play a role in transcription regulation [82]. Unlike other CIA proteins, yeast Cia1 mainly localizes in the nucleus, suggesting that it may also have a transcriptional role like its human counterpart [53,80,83]. Patients with biallelic loss-of-function mutations in CIA1 develop neuromuscular and multisystemic problems, including muscle weakness, respiratory insufficiency, iron deposition in deep brain nuclei and learning difficulties. These mutated CIA1 variants lead to failure in recruiting Fe-S client proteins, thus resulting in compromising the activity of Fe-S proteins, such as DNA helicases, polymerases and repair enzymes that rely on the CIA complex to acquire their Fe-S cofactors [84]. Cia1/CIA1 appears to be required for Fe-S transfer to all known Fe-S target proteins, while it is dispensable for the maturation of the Fe-S cluster-containing CIA proteins, such as Cfd1, Nar1 and Nbp35 in yeast, and CIAPIN1 in mammals [61,85]. Recent experiments have demonstrated that [2Fe-2S]^+^ cluster assembly on the target proteins Aft1-Aft2 and Yap5 in yeast, and aldehyde oxidase in mammals, occurs in a CIA-independent manner, thus suggesting that the role of Cia1/CIA1 in target maturation needs further validation [85].

#### 3.3.2. Cia2

Yeast Cia2 is the homolog of human CIA2B (also named FAM96B or MIP18). In addition to CIA2B, human cells express a Cia2-like protein (CIA2A) that is particularly required for the maturation of key regulators of iron homeostasis [86,87]. CIA2B is also important in coordinating ATP and Fe-S cluster homeostasis during mitosis. In fact, it functions as a scaffold protein that recruits Creatin Kinase B and other components of the MMXD complex, which is formed by MMS19, CIA2B and the helicase XPD, and it is involved in chromosome segregation [88,89].

In yeast, Cia2 is the central component of the targeting complex, tethering Met18 to Cia1 [79,81]. It is a small acidic protein that contains five conserved motifs distributed between an intrinsically disordered N-terminal domain and C-terminal domain of unknown function 59 (DUF59). The disordered domain is dispensable for the binding of the other subunits of the targeting complex in vitro, but in vivo, the deletion of the N-terminal domain negatively impacts CIA function. In fact, the deletion of the first 102 amino acids in yeast Cia2 (*cia2-Δ102*) is sufficient to support cell viability but causes compromised functionality and a diminished ability to support CIA target maturation. This is possibly due to the lower stability of this variant in vivo [79]. The C-terminal region of Cia2 contains the DUF59 domain (Motif 3 and Motif 4) and about 40 amino acids that form Motif 5. Motif 5 is unique in eukaryotic DUF59 proteins, indicating its potential significance in the formation of the targeting complex [61,79]. In Motif 5, glutamate 208 has been demonstrated to be important for Cia1-Cia2 interaction but not for interaction with Met18 in vitro. Mutations of this residue in glycine or alanine do not impair Cia2 function. However, in vivo, they destabilize the Cia1-Cia2 complex, which in turn affects the amount of the Met18-Cia2 complex that can be detected with biochemical assays. They also cause a diminished activity of cytosolic Fe-S cluster proteins [79,90]. Cysteine 161 in yeast and 93 in humans (Motif 4) is critical for the function of Cia2. Mutations of this cysteine cause cell lethality and dominant-negative effects in yeast. This result suggests that Cia2 may engage in protein complexes that are sequestered by the Cia2-C161A mutant variant, thereby disrupting the activity of the wild-type protein and sequestering Met18 and Cia1 in nonfunctional CIA targeting complexes [79,91]. Similarly, mutations in Motif 3 lead to nonfunctional alleles and slow-growing yeast strains. Surprisingly, mutations in Motif 3 and Motif 4 produce no observable perturbations in Cia2′s interaction with the other components of the CIA targeting complex, suggesting that Cia2 uses the DUF59 domain for an additional, yet-unknown function. It has been suggested that Cia2 could play a cluster-carrying role and could interact with the Fe-S cluster as it is inserted into the target apoproteins [79]. Together, these findings highlight how difficult it can be to understand these molecular details and put together in vitro and in vivo results and suggest that Cia2 is not just a mediator of protein–protein interactions, but it likely has an additional, yet-unidentified function in the final cluster insertion step [79].

#### 3.3.3. Met18

Met18/MMS19 is the biggest protein of the CIA targeting complex. It regulates the maturation of several cytosolic Fe-S proteins [7]. In contrast to all the other known CIA proteins, Met18 is not essential for cell viability in yeast, while *MMS19* knockout is embryonically lethal in mice [61,92]. A lack of Met18/MMS19 is associated with a multitude of cellular phenotypes, including defects in methionine synthesis (yeast only), impaired chromosome segregation, sensitivity to genotoxic agents or elongated telomeres [61]. Met18 is a 118 kDa protein with an alpha solenoid structure composed entirely of HEAT repeats (Huntington, Elongation Factor 3, Protein phosphatase 2A and TOR kinase), each including two alpha helices that are known to mediate protein–protein interactions and are often part of large protein complexes. In humans, MMS19 lysine residues from helices 45 and 46 are responsible for the interaction between MMS19 and CIA2B, while no direct interaction between MMS19 and CIA1 has been detected. In these helices, the ubiquitination of lysine residues is likely to preclude CIA2B binding due to steric clashes. Besides directing MMS19 to the proteasome for degradation, thus shutting down the CIA pathway, ubiquitination could serve as a regulatory mechanism that immediately inhibits Fe-S protein biogenesis and prevents the further transfer of Fe-S clusters into MMS19-dependent client proteins [93].

The C-terminal region of MMS19 interacts with CIA2B, while the rest of the protein is accessible for interactions with client proteins or early-acting factors of the CIA pathway [93]. Structural studies in yeast have shown that Met18 adopts a different conformation from those observed in the human and mouse homologs [93] that allows it to form hexamers and tetramers in the absence of Cia2. This oligomerization may protect Met18 from degradation when the CIA complex is paused between clients and/or protect it from interactions with non-clients or holo-clients. In these conformations, the residues in the C-terminal region that appear to be involved in binding Cia2 (R1010, R1013, R1020), and those in the N-terminal region that bind the client proteins (for example, R144, K187 and F217) are hidden, indicating that Cia2 cannot bind Met18 without disrupting these higher-order oligomeric states. The addition of Cia2 causes these oligomeric states of Met18 to dissociate and form Met18-Cia2 complexes [94]. The precise molecular mechanisms by which Cia2 achieves this feat are still unclear.

### 3.4. Assembly of CIA Targeting Complex and Target Maturation

Met18/MMS19, Cia1 and Cia2/CIA2B form the CIA targeting complex, in which Cia2/CIA2B is the central scaffold that links Met18/MMS19 to Cia1 [81,95]. The yeast CIA complex is formed by the docking of two Cia1-Cia2_2_ complexes with Met18, so it comprises one Met18, four Cia2 and two Cia1 polypeptides [81]. In human cells, the CIA complex exists in a dynamic monomer–dimer equilibrium in solution, but further studies are needed to investigate its exact structure [93].

The CIA targeting complex can form various binary and ternary subcomplexes with distinct target specificities. Different human CIA targeting complex components exhibit a high specificity for the maturation of dedicated Fe-S proteins [7]. In fact, proteomic studies show the differential binding of CIA1, CIA2B and MMS19 to the collection of cytosolic and nuclear Fe-S proteins. The ternary CIA targeting complex appears to have the broadest substrate spectrum, mediating the maturation of numerous Fe-S proteins such as dihydropyrimidine dehydrogenase (DPYD), which is involved in nucleotide production, the ATP-binding cassette protein ABCE1 and the helicase XPD. MMS19 plays only a minor role in the maturation of the nucleotide metabolism protein glutamine phosphoribosylpyrophosphate amidotransferase (GPAT), while CIA2B is not crucially required for DNA Pol δ. CIA2A was instead proposed to participate in the regulation of iron homeostasis in human cells, modulating the activity of the iron regulatory proteins IRP1 and IRP2. CIA2A seems to stabilize IRP2 and contribute to the maturation of IRP1, even if no direct interaction has been detected so far between CIA2A and IRP1 [7,86]. The formation of subcomplexes with different specificities appears to be a peculiarity of human cells. However, recent data showed that CIA substrate recognition can be performed by either the full targeting complex or its stable subcomplexes in yeast. For example, Met18-Cia2 can bind the client protein Rad3, whereas the binding of Leu1 requires the full targeting complex [81].

CIA proteins deliver Fe-S clusters to more than thirty thus-far-known client proteins which are structurally distinct. This raises a fundamental question: how do the same three CIA proteins recognize clients of different sizes and shapes while avoiding holo-clients and non-clients [94]? The only similarity detected in different apo-targets is represented by the cysteine residues required to bind the [4Fe-4S]^2+^ cofactors. However, the CIA complex does not recognize any Fe-S protein in vitro. In fact, it cannot interact with bacterial Fe-S proteins, demonstrating that the cluster-binding cysteine motif is not sufficient for target recognition [81].

Since different CIA subcomplexes interact with distinct subsets of targets, each subunit of the targeting complex contributes to target recognition [81]. Structural analysis of the human CIA complex has revealed the presence of two key interaction sites for recognizing client proteins. One is a high-affinity binding site on CIA1, likely common to all client proteins, while the second is located within the N-terminal domain of MMS19. The latter site facilitates the recruitment of MMS19 client proteins by increasing their binding affinity due to cooperativity between the two sites. The plasticity of MMS19 enables the complex to adapt to diverse molecular shapes of client proteins: for example, MMS19 adopts a different conformation based on whether the CIA complex is bound to DNA2, which is more globular, or to a more elongated protein such as DNA primase. The significant conformational plasticity of MMS19 not only aids in client protein recognition but may also play a role in the mechanism of Fe-S cluster transfer from CIA2B into the targets. Cryo-EM studies with the client protein primase have shown that when DNA primase is bound to the complex, the structure is not in a relevant state for Fe-S cluster transfer, as the primase binding site is distant from the predicted cluster-binding site on CIA2B. This indicates that a conformational change or other factors may be required to facilitate Fe-S cluster transfer from the CIA complex to its client proteins. Complex plasticity may facilitate the transfer of Fe-S clusters, bringing CIA2B closer to the client protein. Alternatively, additional factors might be required to bridge CIA2B and the client proteins during Fe–S cluster transfer [93,94]. Also, in yeast, Met18′s conformational flexibility is key for the recognition of clients of various sizes and shapes because its curvature adapts to its binding partners. Together, these findings suggest that multiple states of the CIA complex exist: a state in which the Fe-S cluster is delivered by Nar1, a state in which the cluster is bound to Cia2 and another in which the cluster is bound to the target protein [94]. Further studies are needed to clarify the structure and activity of these different complexes.

## 4. Role of CIA Machinery in Response to Replication Stress

The CIA targeting complex is important to safeguard DNA replication under stress conditions. In fact, in both yeast and mammals, Met18/MMS19 defects result in increased sensitivity to the alkylating agent methyl methanesulfonate (MMS) and to UV light, which interfere with DNA replication, but not to X-ray irradiation [96,97,98,99]. Furthermore, yeast cells expressing the Cia2-Δ102 variant are viable but hypersensitive to both MMS and hydroxyurea (HU), which reduces the dNTP pool, suggesting that the Cia2 N-terminal domain is required for the maturation of Fe-S proteins involved in the replication stress response [79].

The hypersensitivity to replication stress of cells with CIA defects could be due to checkpoint defects, as suggested by both the reduced Rad53 phosphorylation observed in HU-treated *met18Δ* yeast cells and the defective CHK1 phosphorylation detected in HU-treated human HeLa cells [92]. Other studies indicate instead that CIA is not required for checkpoint activation. In fact, the phosphorylation of checkpoint targets Rad53 and CHK1 was observed in both MMS-treated yeast cells lacking Met18 and HU-treated HEK293 human cells with MMS19 depletion, and these phosphorylations were enhanced in CIA-deficient cells compared to the wild type, suggesting that the checkpoint could be hyperactivated in the absence of CIA functions [99,100]. HU treatment causes a prolonged G1 phase and delayed S phase entry in cells with Met18/MMS19 deficiency [92]. Since checkpoint signals in HU are generated during ongoing DNA replication, the reduced checkpoint observed in cells treated in G1 with HU could merely be a consequence of the persistence of these cells in G1. Overall, these findings suggest that the Fe-S machinery is not essential for checkpoint activation but instead facilitates the cellular response to replication stress by minimizing the accumulation of structures that trigger checkpoint activation. In doing so, it enhances the efficiency of the checkpoint pathway in managing replication stress. This hypothesis is supported by the finding that CIA inactivation creates a dependency on the Mec1/ATR pathway for cell survival. In yeast, *MET18* deletion reduced the viability of cells carrying a *mec1-4* temperature-sensitive allele, although the underlying mechanism was not further investigated [101]. Similarly, the depletion of either *MMS19* or *CIA2B* sensitizes triple-negative breast cancer cells to treatment with inhibitors of ATR or CHK1 proteins [100]. Therefore, Met18/MMS19 supports the functions of the Mec1/ATR pathway in DNA replication control, and this function is shared by other CIA components, suggesting that the CIA complex supports cell viability in the absence of the replication checkpoint by promoting the maturation of Fe-S target proteins.

The Mec1/ATR pathway supports cell viability when DNA replication is threatened by a perturbation of DNA polymerase activity, DNA damage or poor or insufficient availability of dNTPs, but it is also crucial for DNA replication in untreated conditions [102,103]. Recent findings indicate that in yeast, Mec1 is transiently activated every time cells enter S phase in short regions centered on early replication origins, reflecting the activation observed under replication stress [104]. This transient activation of Mec1 in unperturbed S phase is suppressed by increasing the dNTP pool before cells enter S phase, indicating that the scarce availability of dNTPs when DNA replication starts creates a local replication stress that activates Mec1, which in turn stimulates RNR activity and concomitantly prevents replication fork collapse [104]. The CIA complex takes part in the regulation of these processes, likely acting independently of the ATR/Chk1 pathway and supporting the functions of Mec1/ATR both in untreated conditions and under replication stress. Below, we describe how the Fe-S machinery can support the replication stress response by controlling DNA polymerases stability, the recovery of stalled replication forks and the availability of dNTPs. We suggest that the regulation of these aspects by the Fe-S machinery contributes to supporting faithful DNA replication under both normal and stressed conditions (Figure 3).

### 4.1. CIA and the Control of DNA Polymerases

A clear connection between Fe-S clusters and DNA polymerases in both yeast and humans was made in the first decade of the 2000s, beginning with the identification and characterization of the yeast *pol3-13* mutant allele [105]. This allele presents a mutation in the last cysteine within the Pol3 CysB motif and a synthetic lethal phenotype in combination with mutations in both CIA components Met18, Nbp53, Dre2 and Tah18 and checkpoint factors, such as Rad53 [105]. Subsequent works identified Fe-S clusters in both yeast and mammalian DNA primase [15,106]. Since then, the CIA complex has been documented to transfer Fe-S clusters to the Fe-S center of DNA polymerases and primases involved in bulk DNA replication as well as in translesion DNA synthesis, thus regulating their stability and/or their activity [107,108]. However, exploring in vivo the consequences of Fe-S impairment in DNA polymerases remains difficult, as the insertion of the clusters in the replicative DNA Pol α, ε and δ is essential for cell viability.

Replicative polymerases (α, δ and ε), as well as the DNA primase Pri2/p58, contain Fe-S clusters in their CTD. Both in yeast and in humans, mutations of iron-coordinating cysteines in the primase CTD resulted in the disruption of protein folding, indicating that the clusters play a critical structural role [107]. The Pol2 subunit of polymerase ε contains two [4Fe-4S]^2+^ clusters, one located in the CTD domain, which is stabilized by interaction with the Dpb2 subunit, and the other within the polymerase domain [109,110]. Mutations in the CysX motif, which coordinates the Fe-S cluster in the N-terminal region of Pol2, impair the DNA binding of the complex, thus abrogating both polymerase and exonuclease activities [111].

In addition to increasing protein stability, the Fe-S cluster has been proposed to act as a redox switch that modulates interactions between DNA polymerases and DNA (reviewed in [112]). [4Fe-4S]^2+^ clusters can be oxidized to the [4Fe-4S]^3+^ state or reduced to the [4Fe-4S]^+^ state. Oxidized Fe-S clusters enhance the DNA-binding affinity of their cognate proteins compared to their reduced counterparts. The redox state of the Fe-S center in DNA primase acts as a molecular switch, regulating DNA binding, the interaction of Pol α-primase complex with the RNA primer and the polymerase handoff [113]. The redox state of the Fe-S cluster in the Pol3 CTD instead regulates the rate of DNA synthesis carried out by Pol3 in vitro [114]. Also, the Fe-S cluster of yeast DNA polymerase ε is redox-active, with protein oxidation causing a sharp decrease in DNA synthesis polymerization [115]. The redox switch between oxidized and reduced states can be activated using DNA charge transport chains and can offer a way to slow down or block DNA synthesis in response to oxidative stress [112]. Recent reports demonstrated that besides reducing RNR activity, HU treatment in yeast cells increases the accumulation of reactive oxygen species (ROS), and that in both yeast and human cells ROS cause the disassembly of polymerase complexes into their subunits, thus inhibiting both DNA binding and polymerase activity [116,117]. Importantly, eukaryotic DNA polymerase complexes that contain Fe-S clusters are inhibited by ROS, while prokaryotic DNA polymerases lacking Fe-S clusters are not, suggesting that ROS inhibit polymerase activity by inducing the oxidation of their Fe-S clusters [117]. Therefore, the redox state could modulate the activity of polymerases through the modulation of the Fe-S center.

Mutations affecting Fe-S binding to the Pol3 CTD interfere with Pol δ complex formation and cause cell lethality at high temperatures, suggesting that the Fe-S cluster supports Pol δ structure and function in non-optimal growth conditions [15,105,118]. Furthermore, a mutation in the Pol3 cysteine-rich domain CysB triggers checkpoint activation and increases both DNA damage and genome instability, also in unperturbed growth conditions [119]. The overexpression of *RAD51* and *RAD52* partially suppresses these defects and increases the viability of *pol3-13* mutant cells, indicating that defects in the Pol3 Fe-S center impairs DNA replication and increases DNA damage in S phase, which requires recombination-mediated repair to be fixed [119]. This model is supported by the finding that Pol3 mutations that prevent Fe-S cluster assembly cause a mutator phenotype and the accumulation of complex mutations and small deletions flanked by short homologies, which are compatible with enhanced DNA polymerase slippage and/or template switching events [120]. Importantly, both the rates and the spectra of mutations observed in these *pol3* mutants resemble those observed in the absence of Met18, while they are not caused by mutations that decrease Fe-S addition to Pol ε [120]. Consistently, both in yeast and in mammals, Pol δ interacts with Met18/MMS19, as does DNA primase [92]. Furthermore, Met18/MMS19 inactivation severely affects the stability of both Pol α and Pol δ complexes [92]. Therefore, Met18 appears to stimulate the fidelity of DNA synthesis mainly via the regulation of Pol δ processivity.

As the stability of the replisome is important for replication recovery under replication stress conditions and Fe-S clusters stabilize DNA polymerase complexes and their interactions with DNA, CIA can support DNA replication under stress conditions at least in part by stabilizing DNA polymerases at stalled replication forks (Figure 3).

### 4.2. CIA and the Restart of Replication Forks

Recombination is essential for restoring active replication forks under replication stress and during replication recovery. Additionally, helicases and nucleases play a key role in fork remodeling and the recombination-dependent recovery of stalled forks, contributing significantly to replication fidelity and genome stability [24]. DNA helicases and nucleases are key targets of the CIA targeting complex, and their regulation through Fe-S cluster insertion may facilitate DNA replication recovery under stress conditions. However, the precise roles of the CIA complex and Fe-S clusters in this process remain largely unexplored. The CIA targeting complex inserts Fe-S clusters in the nuclease/helicase DNA2, and the helicases Rad3/XPD, FANCJ, RTEL1 and Chl1/CHLR1/DDX11, which are involved in different mechanisms of DNA repair and in the cohesion of sister chromatids [99,121]. Although both DNA repair and cohesion are essential to maintain genome stability during DNA replication, Dna2 is likely the Fe-S cluster protein mainly involved in fork recovery under replication stress (Figure 3). In yeast, Dna2 contributes to Mec1 activation at stalled replication forks [29]. Furthermore, DNA2 depletion leads to the accumulation of toxic ssDNA structures behind the replication forks that activate the DNA damage response [122]. Finally, Dna2 has been shown to counteract the accumulation of reversed forks and to promote replication fork restart both in yeast and in mammals [123,124]. The contribution of the CIA targeting complex to regulating these Dna2 functions has not been completely defined. However, as the checkpoint is activated in cells lacking Met18 [99,100], Fe-S cluster addition to Dna2 is likely not required for the Dna2-dependent activation of Mec1.

The Dna2 nuclease domain contains four cysteine residues that coordinate a [4Fe-4S]^2+^ cluster. The coordination of this cluster in DNA2 is unusual, as one of the cysteine residues that binds the cluster is located several hundred amino acids upstream of the other three [125,126]. Dna2 mutations that impair Fe-S assembly in yeast do not affect protein stability or the ability of Dna2 to bind DNA, although they impact the way in which the protein binds DNA, and both Dna2 nuclease and ATPase activities [126]. The loss of the Fe-S cluster in yeast Dna2 also does not impact Dna2 binding to RPA, which recruits Dna2 to 5′-ended ssDNA [126]. The crystal structure of mouse Dna2 indicates that the Fe-S cluster stabilizes a crossover loop within the nuclease domain, forming the base of a cylinder through which ssDNA is channeled [127]. This DNA-binding tunnel undergoes a conformational change following Fe-S cluster addition, which likely facilitates the access of DNA to the nuclease domain. However, Fe-S is not absolutely required for DNA binding, as DNA2 mutant variants carrying mutations in the Fe-S center are still able to bind DNA in vitro. Even in conditions where the protein could bind DNA, the absence of the Fe-S cluster severely impairs the nuclease, helicase and ATPase activities of human DNA2, indicating that the Fe-S cluster specifically affects these Dna2 functions [128]. Fe-S cluster mutants have defects in DNA replication and repair in vivo that correlate in intensity with their effect on catalytic activities in vitro [126]. However, whether and how the Fe-S cluster impacts Dna2 functions at replication forks are still open questions.

Both in yeast and in mammals, Chl1/ChlR1/DDX11 is an ATP-dependent Fe-S cluster DNA helicase required for sister chromatid cohesion. In mammals, the inactivation of the ChlR1/DDX11 gene causes replication stress, renders cells dependent upon DNA damage response genes for survival and causes sensitivity to MMS, mitomycin and camptothecin [129]. *CHLR1/DDX11* depletion was also found to sensitize cancer cells to poly ADP-ribose polymerase (PARP) inhibitors and platinum-based chemotherapy and enhances the sensitivity to chemotherapy of cells with *BRCA1* or *BRCA2* mutations [130]. ChlR1/DDX11 interacts physically with the replication fork component TIMELESS to assist replisome progression [131,132] and promotes ssDNA generation at DNA lesions, thus stimulating both recombination-mediated DNA repair and CHK1 activation [130,133]. These findings suggest that Chl1/ChlR1/DDX11 can take part in replication fork recovery under replication stress (Figure 3). However, how this helicase promotes fork recovery and whether the Fe-S cluster is important for this function are still unknown.

Finally, the Fe-S-mediated regulation of Pol ε could also be involved in replication fork recovery. A switch between polymerase/exonuclease activities of Pol ε was recently documented. This switch was regulated by the checkpoint and was proposed to be important to prevent the extensive processing of stalled forks and promote replication restart [134]. It will be interesting to evaluate whether the Fe-S cluster takes part in this regulation.

### 4.3. CIA and the Control of dNTPs

The CIA targeting complex can support the cellular response to replication stress by regulating the synthesis of dNTPs (Figure 3). In fact, the depletion of *MMS19* results in the deregulation of several metabolites involved in nucleotide production [100]. In both yeast and mammals, CIA inactivation also increases the lethality of cells with replication checkpoint dysfunctions [100,101]. As the replication checkpoint promotes dNTP synthesis [33], the lethality of cells with dysfunctions in both CIA and checkpoint pathways can be due to insufficient or unbalanced dNTP levels [100]. If CIA stimulates dNTP production, CIA inactivation can also exacerbate the reduction in dNTP levels caused by HU treatment. This extreme low availability of dNTPs, which is not sufficient to support DNA replication initiation, can be the cause of the delayed G1/S transition observed in HU-treated *met18Δ* yeast cells [92].

How can the CIA complex regulate dNTP levels? Iron metabolism impacts dNTP production because RNR is an iron-dependent enzyme. The catalytic process that converts rNDPs to dNTPs requires tight cooperation between both R1 and R2 RNR subunits and a diferric–tyrosyl radical cofactor (Fe^3+^_2_-Y∙), which is part of the peptide β in the R2 subunit in both yeast and mammals (Figure 4). This cofactor stabilizes a tyrosyl radical, which is essential for the catalytic process. Substrate binding to R1 initiates an electron transfer chain that involves amino acid residues located in both R1 and R2 subunits and leads to rNDP reduction. After catalysis, the RNR enzyme should be regenerated by the reduction of oxidized cysteines by thioredoxin or glutaredoxin [33,34]. Of the two different β peptides in yeast, only the Rnr2 subunit contains the Fe^3+^_2_-Y∙cofactor, while Rnr4 contributes to the assembly of Fe^3+^_2_-Y∙ into Rnr2 [135].

The assembly of the Fe^3+^_2_-Y β cofactor requires a source of iron and a reducing equivalent. In yeast, the glutaredoxins Grx3 and Grx4, together with the electron donor complex Dre2-Tah18, deliver the iron ion in its reduced state to Rnr2 for the formation of the Fe^3+^_2_-Y∙cofactor, supporting RNR activity (Figure 4) [135,136]. Indeed, Grx4 depletion reduces RNR activity due to the inefficient incorporation of iron [136]. Furthermore, Dre2 depletion diminishes both the levels of tyrosyl radical Y∙ and RNR activity, suggesting that Dre2-Tah18 helps in providing reducing equivalents to deliver iron to Rnr2 in its reduced state but is dispensable for iron loading (Figure 4) [135]. Dre2 contains both a [2Fe-2S]^+^ cluster and a [4Fe-4S]^2+^ cluster, and participates in the CIA pathway (Figure 2) [74,137]. Like many other Fe-S assembly pathway proteins, Dre2 is essential for cell viability. The contribution of Fe-S assembly pathway components in the regulation of RNR activity was investigated by expressing genes encoding for these components under the control of the *GAL1* promoter and by shifting the cells from a galactose-containing medium to a glucose-containing medium. The *GAL1*-mediated depletion of Dre2 reduces both the activity and the levels of Rnr2 and Rnr4 subunits in yeast [135]. The low levels of Rnr2 and Rnr4 in Dre2-depleted cells can be related to the regulation of *RNR2* and *RNR4* mRNA stability. In fact, Dre2 inactivation results in the increased transcription of *CTH1* and *CTH2* genes, which encode two RNA-binding proteins that bind specific AU-rich elements in the mRNA 3′ UTR and induce mRNA degradation [138]. The decrease in Rnr2/Rnr4 levels caused by mRNA degradation is at least in part compensated for by an induction of *RNR2* and *RNR4* gene transcription in the absence of Dre2. This increased transcription depends on the removal of *RNR2*/*RNR4* transcriptional repressor Crt1 because of checkpoint activation in Dre2-depleted cells (Figure 4) [138].

The depletion of the ISC components Nfs1 and Atm1 and of the early-CIA components Cfd1, Nbp35 and Nar1 was also found to reduce RNR activity, while the depletion of the late-CIA factor Cia1 did not [139]. In addition, the depletion of Nfs1, Atm1 and Grx3/Grx4 decreases iron loading into Rnr2, while the depletion of Dre2, Cfd1, Nbp35 and Nar1 does not [139]. These results have led to the model in which iron loading into RNR requires ISC but not CIA, with the pathways for RNR and Fe-S cluster biogenesis bifurcating after Dre2-Tah18 [139]. Importantly, the maturation of Dre2 with Fe-S clusters that is necessary for Dre2 functions in vivo—including RNR regulation—depends on ISC but not on CIA [74,137]. Therefore, it appears unlikely that the late-CIA complex could modulate dNTP production through the addition of Fe-S clusters to Dre2.

Despite iron being essential for RNR activity, in both yeast and mammals, iron deficiencies result in a slight increase in the dNTP pool, suggesting that cells possess mechanisms maintaining RNR function also when iron becomes insufficient [140,141]. One of these mechanisms involves an R2 redistribution from the nucleus to the cytoplasm in response to low iron, driven by the RNA-binding proteins Cth1 and Cth2 [141]. Although the late CIA appears to be dispensable for iron radical addition to the RNR R2 subunit, it is still not known whether the CIA targeting complex contributes to regulating RNR stability or activity (Figure 3).

The depletion of either MMS19 or CIA2B results in significant changes in the intracellular levels of intermediates in nucleotide synthesis. While the amount of several metabolites is lower in the absence of a functional CIA machinery than in control cells, other central intermediates in de novo purine and pyrimidine synthesis (such as IMP and UMP) are more abundant in cells with MMS19 or CIA2B depletion than in control cells [100]. Studies that aimed to identify CIA targets revealed that CIA promotes Fe-S cluster transfer to proteins involved in nucleotide metabolisms, such as GPAT/PPAT and DPYD, which regulate de novo purine synthesis and pyrimidine catabolism, respectively [86,99,100]. Reduced DYPD protein levels were detected after the depletion of either CIA2B or MMS19 in breast cancer cells [100]. These findings suggest that CIA2B or MMS19 mutations impact nucleotide homeostasis and that an altered or unbalanced dNTP pool may contribute to the sensitivity to replication stress of cells with CIA dysfunctions. How CIA regulates dNTP production and whether and how this function supports the checkpoint-mediated regulation of dNTP availability are open questions that require further investigations.

Experiments in different organisms, spanning from *E. coli* and yeasts to mammalian cells, showed that either unbalanced or balanced accumulation of dNTPs dramatically increases the mutation rates during DNA replication and repair, therefore resulting in cell death or genetic abnormalities (reviewed in [33,34]). Considering that the dysregulation of the dNTP pool impacts genome stability and can sustain carcinogenesis by increasing the mutation rates and generating DNA damage, and that several approaches to treating cancer heavily exploit the effects of dNTP pool unbalance in DNA metabolism [34,35,142], the definition of the CIA-mediated control of nucleotide homeostasis can increase our understanding of mechanisms leading to cancer development and help the development of new targeted therapies.

## 5. Conclusions

The stability of DNA is crucial for the proper functioning of cells, while genome instability is a hallmark of cancer, which fuels carcinogenesis and the development of resistance to therapies [6]. It is increasingly clear that developing personalized therapies for the treatment of specific types of cancer is essential. This is possible if the mutations present in the tumor and their biological effects are well understood. In fact, specific mutations can make a tumor more or less sensitive to a particular type of therapy or to the use of specific inhibitors. The identification of novel potential therapeutic targets for cancer targeted therapies requires a precise understanding of how cells safeguard DNA integrity and how the involved proteins interact. In this work, we summarize the key pathways activated in response to replication stress and their regulation by Fe-S clusters, which are emerging as crucial cofactors in genome stability maintenance. However, several aspects remain unclear and require further investigation—for instance, how Fe-S clusters modulate helicase and polymerase activity during replication fork recovery, and the role of the CIA complex in regulating dNTP production.

Both replication stress and Fe-S cluster biogenesis genes have been found to be mutated in both tumors and genetic diseases [57,143,144]. In addition, iron metabolism is dysregulated in cancer cells, facilitating tumor initiation, growth and metastasis [145]. Targeting iron metabolism and modulating Fe-S clusters represent promising therapeutic strategies for cancer treatment. Current approaches include iron chelators, which restrict iron availability to tumor cells, and ferroptosis inducers, which trigger cell death through iron-dependent pathways. These therapies are being explored both as monotherapies and in combination with other cancer treatments to enhance efficacy and exploit tumor-specific vulnerabilities [144,145]. Given the interconnections between replication stress and Fe-S cluster biogenesis, the treatment of cancers harboring mutations in Fe-S cluster biogenesis genes may also benefit from the targeting of key regulators of replication stress. Indeed, it has already been observed that dysfunctions in the CIA complex sensitize triple-negative breast cancer cells to ATR and CHK1 inhibitors [100]. Identifying additional proteins involved in replication stress and Fe-S cluster biogenesis, as well as exploring their interconnections, could open new avenues for targeted therapies.

## Figures and Tables

**Figure 1 cells-14-00442-f001:**
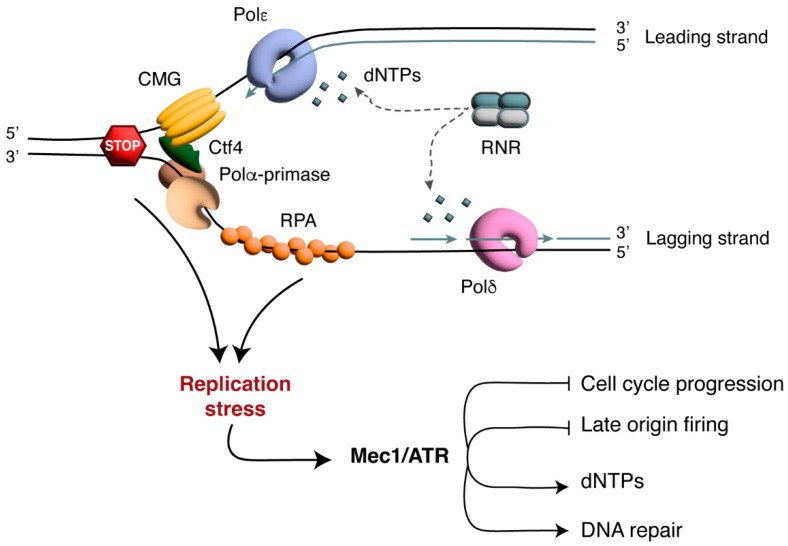
DNA replication and the response to replication stress. The CMG replicative helicase unwinds the parental DNA and allows the recruitment of the Pol α-primase complex through its interaction with Ctf4. The Pol α-primase complex synthesizes RNA primers, which serve as starting points for DNA synthesis. DNA polymerases ε (Pol ε) and δ (Pol δ) catalyze DNA replication on the leading and lagging strands, respectively, by incorporating deoxyribonucleoside triphosphates (dNTPs) produced via the ribonucleotide reductase (RNR)-dependent pathway. When replication fork progression is hindered by obstacles or insufficient dNTP levels, single-stranded DNA (ssDNA) regions are generated and coated by Replication Protein A (RPA). Long RPA-coated ssDNA stretches activate the Mec1/ATR-dependent checkpoint, which delays cell cycle progression, inhibits the firing of late replication origins, upregulates dNTP production and facilitates DNA repair.

**Figure 2 cells-14-00442-f002:**
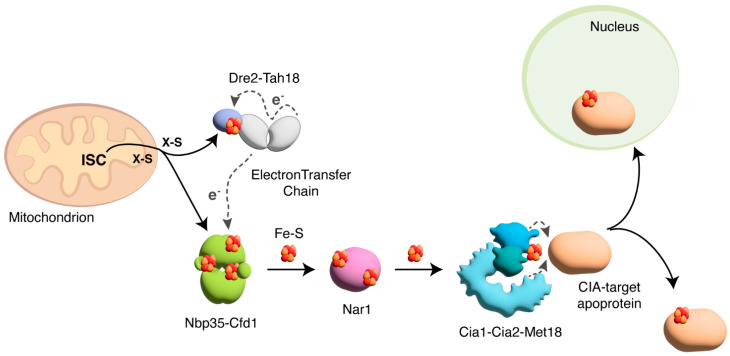
Biogenesis of cytosolic and nuclear Fe-S cluster proteins in yeast. In the mitochondrion, the mitochondrial Fe-S cluster (ISC) system generates a sulfur-containing compound (X-S) and delivers it to the cytosol, where it becomes a substrate of the cytosolic Fe-S cluster assembly (CIA) system. [4Fe-4S]^2+^ clusters are assembled on the Nbp35-Cfd1 heterotetrameric scaffold complex, thanks to the electron transfer chain mediated by Tah18 and the Fe-S protein Dre2. Then, Nar1 promotes the transfer of the Fe-S cluster to the CIA targeting complex Cia1-Cia2-Met18. The CIA targeting complex inserts the cluster into the target cytosolic and nuclear apoproteins.

**Figure 3 cells-14-00442-f003:**
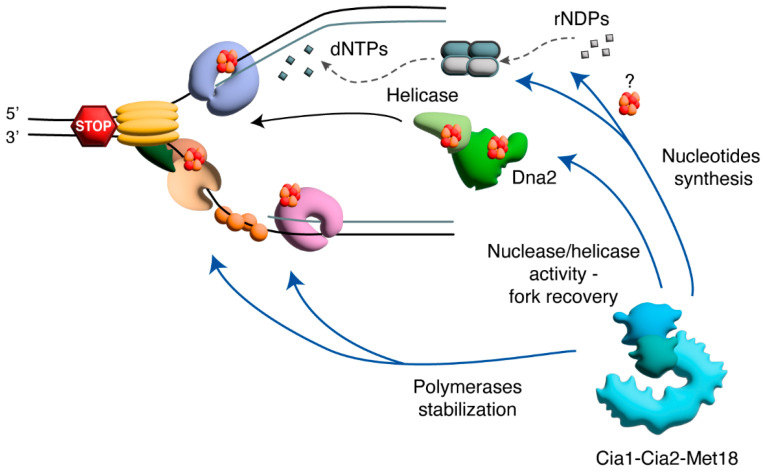
Possible roles of Fe-S cluster proteins in yeast replication stress response. Replication stress causes replication fork stalling. The resumption of DNA replication requires the replisome to be maintained on DNA until the stress is removed, the preservation of the replication fork structure, and a sufficient pool of dNTPs to restart and complete DNA replication. As several Fe-S proteins are involved in these processes, the CIA pathway may contribute to overcoming replication stress by regulating some or all of these factors. Specifically, it could help stabilize DNA polymerases on DNA or enhance the activity of nucleases and helicases, such as Dna2, which participate in fork remodeling and the restoration of a functional replication fork. Finally, CIA appears to regulate the nucleotide pool, although the molecular mechanism underlying this regulation remains unclear.

**Figure 4 cells-14-00442-f004:**
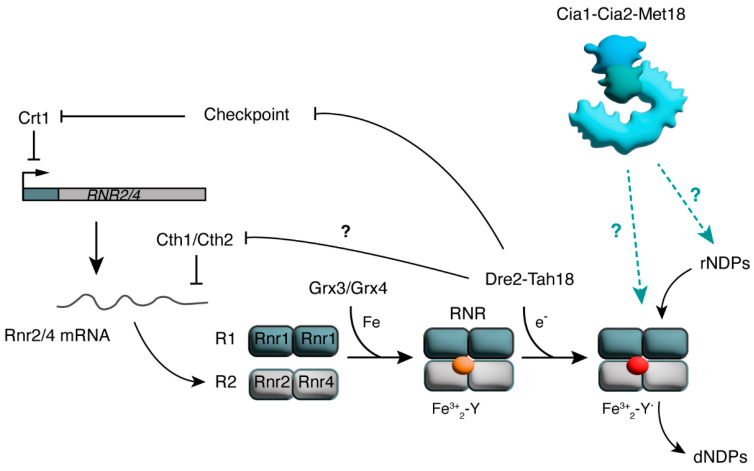
Regulation of RNR by iron in yeast. Active ribonucleotide reductase (RNR) is formed by R1 and R2 subunits. The Rnr2 R2 subunit carries a Fe^3+^_2_-Y∙ cofactor, whose assembly requires the monothiol glutaredoxins Grx3/Grx4 and the electron donor complex Dre2-Tah18. Fe^3+^_2_-Y∙ is necessary for RNR catalytic activity and the conversion of ribonucleoside diphosphates (rNDPs) to deoxyribonucleoside diphosphates (dNDPs). Dre2-Tah18 also controls *RNR2/RNR4* expression both by counteracting checkpoint activation and the subsequent removal of the transcriptional repressor Crt1, and by reducing the levels of the RNA-binding proteins Cth1/Cth2, which in turn regulate *RNR2/RNR4* mRNA stability. Also, the late-CIA complex Cia1-Cia2-Met18 likely participates in the control of dNTP homeostasis, although whether it controls RNR or other steps in dNTP biosynthesis remains to be elucidated.

**Table 1 cells-14-00442-t001:** The Fe-S assembly machinery in yeast and human cells.

	Yeast	Human	Function
ISC system components	Isu	ISCU	Major scaffold protein for Fe-S cluster assembly
	Nfs1	NFS1	Cysteine desulfurase that provides sulfur to Isu/ISCU protein
	Yah1	FDX2	Ferredoxin involved in electron transport chain from NADH to sulfur
	Arh1	FDXR	Ferredoxin reductase that participates in electron transport chain from NADH to sulfur
	Yfh1	Frataxin	Promotes the delivery of iron to Isu/ISCU protein
	Ssq1	HSPA9	Chaperone that stabilizes scaffold protein and induces cluster release
	Jac1	HSC20	Chaperone that stabilizes scaffold protein and induces cluster release
	Isa1	ISCA1	Synthesizes [4Fe-4S]^+^ cluster from [2Fe-2S]^2+^ cluster
	Isa2	ISCA2	Synthesizes [4Fe-4S]^+^ cluster from [2Fe-2S]^2+^ cluster
	Iba57	IBA57	Synthesizes [4Fe-4S]^+^ cluster from [2Fe-2S]^2+^ cluster
	Nfu1	NFU1	Ensures transfer of Fe-S cluster to mitochondrial targets
	Atm1	ABCB7	ABC transporter that exports X-S from mitochondria
CIA system components	Cfd1	CFD1/NUBP2	Forms complex with Nb35/NBP35 that coordinates [4Fe-4S]^2+^ clusters
	Nbp35	NBP35/NUBP1	In complex with Cfd1/CFD1, coordinates [4Fe-4S]^2+^ clusters
	Tah18	NDOR1	Component of cytosolic electron transport chain
	Dre2	CIAPIN	Component of cytosolic electron transport chain
	Grx3-Grx4	PICOT	Promotes Fe-S cluster assembly on Dre2
	Nar1	IOP1	Adapter between early- and late-acting CIA
	Cia1	CIA1	Inserts Fe-S cluster into target proteins
	Cia2	CIA2B/FAM96B/MIP18	Inserts Fe-S cluster into target proteins
	Met18	MMS19	Inserts Fe-S cluster into target proteins

## Data Availability

Not applicable.

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
