# Peer review of "Control of Replication Stress Response by Cytosolic Fe-S Cluster Assembly (CIA) Machinery"

_cells, 2025, doi:10.3390/cells14060442_

Round 1
Reviewer 1 Report
Comments and Suggestions for Authors
General Comments
Overall: The paper provides an overview of the role of Fe-S clusters and the CIA machinery in the replication stress response. It is well-structured and clearly written, with a valuable focus on both yeast and mammalian systems.
Focus: The abstract and introduction effectively define the scope and significance of the review.
Suggestions for Improvement:
- The importance of conserved LYR-like motifs in predicting Fe-S proteins should be addressed.
- The claim that Fe-S clusters serve solely a structural role is highly debatable. They also play essential enzymatic and regulatory functions, with structural stabilization being a secondary role.
- The direct dependence of the cytoplasmic Fe-S assembly pathway on mitochondria for X-S export lacks strong experimental support. Given that key components of the de novo Fe-S machinery (e.g., NFS1, LYRM4) are present in the cytosol, the possibility of a parallel, independent pathway in multicellular eukaryotes should be discussed.
- Frataxin is now recognized as an allosteric effector in Fe-S cluster biogenesis, rather than a direct iron donor. This distinction should be clarified.
- The description of the conversion from rhombic to cubane Fe-S clusters (page 7, lines 317-319) is oversimplified and should be revised for accuracy.
- The conclusion that all cytosolic Fe-S proteins—except Cfd1, Nar1, and Nbp35—receive their clusters from CIAO1 is too definitive. Recent findings suggest this may not always be the case, and as more Fe-S proteins are identified, the reliance on CIAO1 needs further validation.
- The statement on lines 357-358 requires a supporting reference.
- Lines 410-412 should be corrected to reflect that MMS19 knockout is embryonically lethal.
- The role of CIA2A in IRP1 maturation should be presented more cautiously, as the cited study lacks direct evidence.
- Lines 474-476: The number of known CIA complex client proteins is still evolving. The statement should be revised to: " CIA proteins deliver…to more than 30 thus far known client proteins "
Typos
Check the text for typos(examples):
- Page 1, Line 39: "deoxyribonucleotides triphosphate (dNTPs)" → "deoxyribonucleotide triphosphates (dNTPs)."
- Page 2, Line 74: "synthetizes RNA-DNA primers" → "synthesizes RNA-DNA primers."
- Page 3, Line 134: "Rad17 e Mec3" → "Rad17 and Mec3."
Additional Suggestions:
- Ensure all figures are properly cited in the text and that their legends are clear and concise.
Author Response
Comment 1: The importance of conserved LYR-like motifs in predicting Fe-S proteins should be addressed.
Response 1: We agree. We added a brief description of the LYR-like motifs and their importance in Fe-S protein prediction.
Comment 2: The claim that Fe-S clusters serve solely a structural role is highly debatable. They also play essential enzymatic and regulatory functions, with structural stabilization being a secondary role.
Response 2: We agree. We modified the text.
Comment 3: The direct dependence of the cytoplasmic Fe-S assembly pathway on mitochondria for X-S export lacks strong experimental support. Given that key components of the de novo Fe-S machinery (e.g., NFS1, LYRM4) are present in the cytosol, the possibility of a parallel, independent pathway in multicellular eukaryotes should be discussed.
Response 3: We agree. We discussed the possibility of a de novo cytosolic pathway of Fe-S biosynthesis.
Comment 4: Frataxin is now recognized as an allosteric effector in Fe-S cluster biogenesis, rather than a direct iron donor. This distinction should be clarified.
Response 4: We agree. We modified the text.
Comment 5: The description of the conversion from rhombic to cubane Fe-S clusters (page 7, lines 317-319) is oversimplified and should be revised for accuracy.
Response 5: We agree. We modified the text.
Comment 6: The conclusion that all cytosolic Fe-S proteins—except Cfd1, Nar1, and Nbp35—receive their clusters from CIAO1 is too definitive. Recent findings suggest this may not always be the case, and as more Fe-S proteins are identified, the reliance on CIAO1 needs further validation.
Response 6: We agree. We modified the text and discussed the possibility of the existence of a CIAO1-independent pathway of Fe-S protein assembly.
Comment 7: The statement on lines 357-358 requires a supporting reference.
Response 7: We agree. We added the reference.
Comment 8: Lines 410-412 should be corrected to reflect that MMS19 knockout is embryonically lethal.
Response 8: We agree. We corrected the sentence.
Comment 9: The role of CIA2A in IRP1 maturation should be presented more cautiously, as the cited study lacks direct evidence.
Response 9: We agree. We modified the text.
Comment 10: Lines 474-476: The number of known CIA complex client proteins is still evolving. The statement should be revised to: " CIA proteins deliver…to more than 30 thus far known client proteins "
Response 10: We agree. We modified the sentence.
Comment on Typos and Figures: Check the text for typos. Ensure all figures are properly cited in the text and that their legends are clear and concise.
Response: We apologize for typos. We carefully revised the text. We also revised the legends and citations of the figures in the text.
Reviewer 2 Report
Comments and Suggestions for Authors
The work "Control of replication stress response by the cytosolic Fe-S cluster assembly (CIA) machinery" describes the state of the art regarding the regulatory cascades of the replication stress response that depend on Fe-S complexes. It is well written, sufficiently exhaustive, and in-depth, and it describes some current theories for the different regulatory pathways.
The work could be substantially improved by a more detailed description of the components of the described pathways, for example, Nbp35 is mentioned for its importance in the formation of complexes, but it is not defined who it is or where it comes from, although it is correctly referenced in the bibliography; therefore it would be convenient to have a brief definition of this and other elements that are mentioned, perhaps a table could be an option. On the other hand, there are very long paragraphs of descriptions of complexes, as in the case of section 4.3. CIA and the control of dNTPs, in which one or two brief diagrams would help the reader's understanding.
Author Response
Comment 1: The work could be substantially improved by a more detailed description of the components of the described pathways, for example, Nbp35 is mentioned for its importance in the formation of complexes, but it is not defined who it is or where it comes from, although it is correctly referenced in the bibliography; therefore it would be convenient to have a brief definition of this and other elements that are mentioned, perhaps a table could be an option.
Response 1: We improved the description of Fe-S machinery components in the text. We also added a table (Table 1) summarizing the components of the Fe-S assembly machinery in yeast and human cells.
Comment 2: there are very long paragraphs of descriptions of complexes, as in the case of section 4.3. CIA and the control of dNTPs, in which one or two brief diagrams would help the reader's understanding.
Response 2: In the new Table 1 we summarized the components of the Fe-S assembly machinery in yeast and human cells. Furthermore, we added a new figure (Figure 4), summarizing the regulation of RNR assembly and activity by the iron-dependent pathways described in the section 4.3.